# An In-Depth Characterization of SARS-CoV-2 Omicron Lineages and Clinical Presentation in Adult Population Distinguished by Immune Status

**DOI:** 10.3390/v17040540

**Published:** 2025-04-08

**Authors:** Greta Marchegiani, Luca Carioti, Luigi Coppola, Marco Iannetta, Leonardo Alborghetti, Vincenzo Malagnino, Livia Benedetti, Maria Mercedes Santoro, Massimo Andreoni, Loredana Sarmati, Claudia Alteri, Francesca Ceccherini-Silberstein, Maria Concetta Bellocchi

**Affiliations:** 1Department of Experimental Medicine, University of Rome Tor Vergata, 00133 Rome, Italy; 2Clinical Infectious Disease, Department of Systems Medicine, University of Rome Tor Vergata, 00133 Rome, Italy; 3Department of Oncology and Hemato-Oncology, University of Milan, 20122 Milan, Italy; 4Clinical Microbiology and Virology Unit, Fond. IRCCS Ca’ Granda Ospedale Maggiore Policlinico, 20122 Milan, Italy

**Keywords:** SARS-CoV-2 Omicron, genetic variability, hospitalization

## Abstract

This retrospective study analyzed SARS-CoV-2 Omicron variability since its emergence, focusing on immunocompromised (IPs) and non-immunocompromised adult people (NIPs). Phylogenetic analysis identified at least five major Omicron lineage groups circulating in Central Italy, from December 2021 to December 2023: (a) BA.1 (34.0%), (b) BA.2 + BA.4 (25.8%), (c) BA.5 + BF (10.8%), (d) BQ + BE + EF (9.2%), and (e) Recombinants (20.2%). The BA.2 + BA.4 lineages were more common in IPs compared to NIPs (30.9% vs. 17.8%, respectively; *p* = 0.011); conversely, Recombinants were less prevalent in IPs than in NIPs (16.0% vs. 27.1%, respectively; *p* = 0.018). High-abundant single nucleotide polymorphisms (SNPs; prevalence ≥ 40%) and non-synonymous SNPs (prevalence ≥ 20%) increased during the emergence of new variants, rising from BA.1 to Recombinants (54 to 92, and 43 to 70, respectively, both *p* < 0.001). Evaluating the genetic variability, 109 SNPs were identified as being involved in significant positive or negative associations in pairs (phi > 0.70, *p* < 0.001), with 19 SNPs associated in 3 distinct clusters (bootstrap > 0.96). Multivariate regression analysis showed that hospitalization was positively associated with one specific cluster, including S686R and A694S in Spike and L221F in Nucleocapsid (AOR: 2.74 [95% CI: 1.13–6.64, *p* = 0.025]), and with increased age (AOR:1.03 [95% CI: 1.00–1.06], *p* = 0.028). Conversely, negative associations with hospitalization were observed for female gender and previous vaccination status (AORs: 0.34 [95% CI: 0.14–0.83], *p* = 0.017 and 0.19 (95% CI: 0.06–0.63, *p* = 0.006, respectively). Interestingly, the S686R SNP located in a furin cleavage site suggests its potential pathogenetic role. The results show how Omicron genetic diversification significantly influences disease severity and hospitalization, together with age, sex, and vaccination status as key factors.

## 1. Introduction

The Omicron variant of the Severe Acute Respiratory Syndrome Coronavirus 2 (SARS-CoV-2) was first identified in South Africa in late 2021 [1]. It quickly evolved, accumulating single nucleotide polymorphisms (SNPs) that generated new variants, characterized by many mutations, particularly in the spike protein [2,3]. Omicron has shown increased transmissibility compared to previous variants, such as Alpha, Beta, and Delta [4]. Omicron lineages, whether pure variants or Recombinants, are currently the only variants circulating worldwide, indicating a significant shift in the pandemic landscape [5]. Several key factors can explain the dominance of the Omicron variant in the current landscape of COVID-19. First, Omicron has demonstrated a remarkable ability to evade acquired immunity from previous infections and vaccinations [6,7]. Individuals who had previously contracted the virus or had been vaccinated can be reinfected by Omicron, contributing to its spread. Additionally, Omicron has spread rapidly due to its high transmissibility [8]; although highly contagious, previous variants, such as Alpha, Beta, and Delta, could not compete with Omicron [9]. These characteristics have allowed Omicron to establish itself as the dominant variant globally, significantly altering the course of the pandemic (https://gisaid.org/, accessed on 10 February 2025).

To better reflect the current variant landscape, which is dominated by Omicron descendent lineages, WHO updated its tracking system and working definitions of the variants of concern (VOCs) and the variants of interest (VOIs) on 15 March 2023 [10]. The WHO added a new category to the variant tracking system, named “Omicron sub-variants under monitoring” (VUMs). To date, among Omicron descendent lineages, one VOI (JN.1) and seven VUMs (KP.2, KP.3, KP.3.1.1, JN.1.18, LB.1, XEC, and LP.8.1) require tracking (https://www.who.int/activities/tracking-SARS-CoV-2-variants, accessed on 10 February 2025).

The Omicron variant primarily reaches cells of the upper respiratory tract using its spike protein, which binds to the ACE2 receptors on the surface of these cells [11]. Once inside the virus hijacks the cellular machinery to replicate itself. This process damages the infected cells and triggers an immune response, leading to the characteristic symptoms of COVID-19, which include cough, fatigue, and flu-like symptoms. In most people, the disease is mild; however, in some individuals, particularly the elderly and people with compromised immune systems, it may progress from pneumonia to severe complications including death [12].

Immunocompromised people (IPs) are generally at higher risk of serious infections, hospitalization, and the development of persistent infections; they include various subpopulations, such as patients with cancer undergoing chemotherapy or immunotherapy, patients with autoimmune diseases, individuals receiving immunosuppressive therapy for solid organ transplants, those with advanced or untreated HIV infection, etc. [13].

In several case reports, it is reported that SARS-CoV-2 infection in IPs is very prolonged, with viral shedding observed for many weeks and infection persistence observable for some months with genomic and subgenomic RNA detected for several months after the initial diagnosis [14,15]. During COVID-19, multi-mutational strains can arise in IPs, which can act as a reservoir for accumulating mutations and the subsequent emergence of new variants. This process may preferentially occur in incomplete immune control where the infection and viral replication are very prolonged [16,17].

In this setting, this study aims to provide a retrospective analysis of the variability of the SARS-CoV-2 Omicron variant during its spread over the first two years in Central Italy. This analysis specifically focused on comparing the impact and progression of the virus in IPs and non-immunocompromised adult people (NIPs).

## 2. Materials and Methods

### 2.1. Population Study

This retrospective study included 306 nasopharyngeal swabs (NSs) that tested positive for SARS-CoV-2 and were collected from adult people attending the University Hospital of Rome Tor Vergata in Central Italy between December 2021 and December 2023. All participants were NS real-time PCR-positive for four specific genes: envelope (E), nucleocapsid (N), and RNA-dependent RNA polymerase/spike (RdRp/S) with cycle threshold (Ct) values < 35. The Ct values were obtained using the Allplex™ SARS-CoV-2 Assay by Seegene, (target E, N, and RdRp/S). The study protocol for the collection of samples and subsequent sequencing of SARS-CoV-2 was approved by the Ethics Committee of Fondazione PTV Policlinico Tor Vergata (register number 46/20, dated 26 March 2020) and conducted following the ethical standards of the 1964 Declaration of Helsinki. All individuals allowed viral sequencing for surveillance and/or research purposes. Anonymous demographic, epidemiological, and clinical data were collected retrospectively in compliance with the European General Data Protection Regulation (EU Regulation 2016/679) and the Italian Legislative Decree 196/2003.

### 2.2. Viral RNA Extraction and Next-Generation Sequencing (NGS)

Viral RNA was extracted from nasopharyngeal swabs with MagPure virus DNA/RNA Purification Kit (Hangzhou Bigfish Bio-tech Co., Ltd., Hangzhou, China) according to the product specifications, by using BIG FISH™ Nuetraction 32/96 Nucleic Acid Purification System. The RNA extracted was used for SARS-CoV-2 whole-genome next-generation sequencing (NGS) by using the COVIDSeq Assay index 1 (Illumina Inc., San Diego, CA, USA) according to the manufacturer’s instructions, and 15 pM of denatured pool was sequenced on a MiSeq instrument (Illumina, San Diego, CA, USA) with 2 × 150 bp paired-end reads using MiSeq Reagent Kit V2 (2 × 150) Illumina Inc.

### 2.3. Bioinformatics Analyses: Definition of Mutational Pattern and Related Clade and Prevalence of Mutations

The quality control of the raw data obtained in the fastq format was performed by Trimmomatic software [18] to remove adapters, PCR primers, and poor-quality reads. Fastq files were analyzed with VirVarSeq software version 1 (https://sourceforge.net/projects/virtools/files/; last accession on 10 February 2025) [19] using the GenBank reference genome NC_045 512.2 (Wuhan, collection date: December 2019) as a reference. For each sample, a consensus sequence with a prevalence cut-off of 2% was generated using quasitools [20], and these were uploaded on nextstrain (https://clades.nextstrain.org, last accession on 10 February 2025) in order to assign the clade [21] and on Pangolin lineages (https://pangolin.cog-uk.io/ last accession on 10 February 2025) to assign the variant [22]. SNP variants were called with freebayes (v1.3.2) [23], and all SNPs having a minimum supporting read frequency of 2% with a depth ≥ 10 reads were retained. Synonymous and non-synonymous SNPs characterizing Omicron lineages were defined as high-abundant if characterized by a mutated read frequency ≥ 40%, Sanger-like if characterized by a mutated read frequency ≥ 20%, and low-abundant if characterized by a mutated read frequency between 5% and 19%.

### 2.4. Phylogenetic Analysis and Estimation of Evolutionary Rate

The consensus sequences obtained of the SARS-CoV-2 whole genome were analyzed using the Pangolin application to identify the lineages. According to phylogenetic analysis, the consensus viral sequences were then grouped into five main lineage groups: (a) BA.1, (b) BA.2 + BA.4, (c) BA.5 + BF, (d) BQ.1 + BE + EF, and (e) Recombinants. Briefly, the sequences were aligned using MAFFT v7.475, followed by manual inspection with BioEdit. The final alignment consisted of 306 sequences and 82 references downloaded from GISAID last accession on 10 February 2025 with each being 29,495 nucleotides in length.

To investigate the phylogenetic structure of the epidemic and the evolutionary rate of the Omicron clade, a maximum likelihood (ML) phylogenetic tree was constructed using IQ-TREE2 (v2.1.3) with 1000 bootstrap replicates. The best-fit nucleotide substitution model, GTR + F + I + R3, was determined using ModelFinder. The ML tree was examined in TempEst to determine the correlation between genetic diversity (root-to-tip divergence) and the time of sample collection.

### 2.5. Covariation Analysis

The strength of covariation among SNPs of SARS-CoV-2 was assessed by calculating the binomial-correlation coefficient (phi) for each pair of SNPs. This analysis included all SARS-CoV-2 SNPs with a prevalence greater than 5% in the overall population. The phi coefficient and corresponding *p*-values for all possible pairwise SNP combinations were computed using a custom script implemented in R software, version 3.4.1. Pairwise associations were considered statistically significant if the *p*-value was <0.001 and the phi was >0.7 and <−0.7. To further investigate the covariation structure of the SNPs, an average linkage hierarchical agglomerative clustering was performed. The statistical robustness of the clustering was validated through bootstrap analysis with 10,000 replications. Clusters with a bootstrap value of 0.70 or higher were considered well supported.

### 2.6. Statistical Analysis

Descriptive statistics were presented as median values and interquartile range [IQR] for continuous variables, and as number (percentage) for categorical variables. To determine significant differences, the Chi-squared test and the Kruskal–Wallis were applied for categorical and continuous variables, respectively. To define differences between IPs and NIPs, Fisher exact test and Mann–Whitney tests were used. A multivariate logistic regression analysis evaluated demographic and virus-related factors associated with disease severity. All statistical analyses were performed using the SPSS software package for Windows (version 23.0, SPSS Inc., Chicago, IL, USA). A two-sided *p*-value of less than 0.05 was considered statistically significant.

## 3. Results

### 3.1. Patients’ Characteristics

A total of 306 adult individuals infected with SARS-CoV-2 were analyzed and characterized. Overall, 50.9% were female with a median age of 64 [IQR: 51–76] years. Among 297 individuals with an available vaccination status, 7.1% reported not being vaccinated against SARS-CoV-2. Regarding the infection severity, 47 (15.4%) individuals needed hospitalization, whereas the majority of individuals were non-hospitalized (n = 259, 84.6%).

Considering the immune status of participants, 188 were IPs and 118 NIPs. Among the 188 IPs, 26 were Hospitalized (HIPs) and 162 were Non-Hospitalized Immunocompromised People (NHIP). While, among the 118 NIPs, 21 were Hospitalized non-immunocompromised People (HP), and 97 were Non-Hospitalized and non-immunocompromised People (NHP).

Significant differences emerged when comparing the characteristics of the groups. In detail, the IPs were younger than NIPs (median, IQR years: 60 [47–71] vs. 73 [60–81], *p* < 0.001). Furthermore, the frequency of vaccination against SARS-CoV-2 was significantly lower among hospitalized individuals compared to those who were non-hospitalized (76.9% vs. 95.3%, respectively; *p* < 0.001). Pneumonia was observed exclusively in hospitalized individuals. When comparing hospitalized IPs vs. NIPs individuals, the prevalence of pneumonia was found to be significantly lower in hospitalized IPs compared to NIPs (34.6% in HIP vs. 85.7% in HP, *p* = 0.002).

In addition, a significant difference was observed when comparing the median Ct values between hospitalized and non-hospitalized individuals, particularly for the E and RdRp/S targets (*p* = 0.035 and *p* = 0.038, respectively). A difference in gender distribution was also observed, with a lower number of females requiring hospitalization, both in the comparison between hospitalized and non-hospitalized immunocompromised females (26% HIP vs. 56.2% NHIP, *p* = 0.006), and more generally between hospitalized and non-hospitalized females, regardless of immunological status (34.0% vs. 54.1%, *p* = 0.012).

The demographic and clinical characteristics of 306 SARS-CoV-2-infected individuals included in the study are shown in Table 1.

### 3.2. Distribution of SARS-CoV-2 Lineages and Phylogenetic Analysis

From December 2021 to December 2023, a total of 306 SARS-CoV-2 whole-genome sequences were collected, identifying 81 distinct Omicron variants following Nextstrain Clades and Pangolin classification (Appendix A). By phylogenetic analysis, the Maximum Likelihood tree constructed with the inclusion of reference sequences, revealed that at least five major Omicron lineage groups were prevalent in our population. The majority of SARS-CoV-2 strains belonged to (a) BA.1 (n = 104, 34.0%), followed by (b) BA.2 + BA.4 (n = 79, 25.8%), (c) BA.5 + BF (n = 33, 10.8%), (d) BQ + BE + EF (n = 28, 9.2%), and (e) Recombinants (n = 62, 20.2%). In detail, within the BA.1 group, BA.1.1 and BA.1.17.2 were the most common variants observed (n= 45 and n= 22, respectively), while in the BA.2 + BA.4 group, BA.2 and BA.2.9 were the most prevalent (n = 37 and n = 17, respectively). In the BA.5 + BF group, BA.5.1 was the most prevalent (n = 7), and in the BQ + BE + EF group, BQ.1.1 was the most prevalent (n = 13). Within the Recombinants group, the most common was the XBB.1.5 (n = 12) and also two sequences of BA.2.86 were included. (Appendix A). The distribution of SARS-CoV-2 sequences against clinical characteristics of individuals is illustrated through a Maximum Likelihood tree in Figure 1.

A different prevalence of the five major Omicron lineage groups appeared when comparing the IPs and NIPs groups. In particular, the BA.2 + BA.4 lineages were more common in IPs compared to NIPs (30.9% vs. 17.8%, respectively; *p* = 0.011). Conversely, Recombinant lineages were less prevalent in IPs than in NIPs (16.0% vs. 27.1%, respectively; *p* = 0.018) (Figure 2).

Looking at potential differences in lineage distributions against the four categories HIP, NHIP, HP, and NHP, the Omicron lineages BA.1 was more predominant in HP than others (57.1% in HP vs. 34.6% in HIP vs. 32.1% in NHIP vs. 32.0% in NHP, *p* = 0.020), while BA.2 + BA.4 lineages were prevalently found in NHIP with respect to others (33.3% in NHIP vs. 15.4% in HIP vs. 14.3% in HP vs. 18.6% in NHP, *p* = 0.001). Furthermore, a slightly increased frequency of BQ + BE + EF lineages was observed in HIP with respect to the others (19.2% in HIP vs. 8.0% in NHIP vs. 9.5% in HP vs. 8.2 in NHP, *p* = 0.062. (Figure 3).

### 3.3. Characterization of SNPs

The observed trend in the prevalence of SNPs across various SARS-CoV-2 Omicron lineages well illustrates the ongoing, continuous evolution of this clade. The analysis revealed an increasing number of high-abundant SNPs (previously defined with a mutated read frequency ≥ 40%) among the different Omicron sublineages. Overall, in the BA.1 lineage, the number of high-abundant SNPs was median [IQR] 54 [49–59], and this number increased to 68 [64–72] in BA.2 + BA.4 and 68 [63–73] in BA.5 + BF, reaching 75 [72–78] in BQ + BE + EF and 92 [84–100] in Recombinants, *p* < 0.001 (Figure 4A). This median increase was most manifest in the Spike protein, where the number of SNPs was 28 [25–31] in BA.1 vs. 29 [28–30] in BA.2 + BA.4 vs. 30 [29–31] in BA.5 + BF vs. 32 [30–34] in BQ + BE + EF vs. 39 [36–42] in Recombinants *p* < 0.001 (Figure 4B). Particularly, in the Receptor Binding Domain (RBD), the median [IQR] was 15 [12–18] in BA.1 vs. 16 [15–17] in BA.2 + BA.4 vs. 17 [16–18] in BA.5 + BF vs. 20 [20–20] in BQ + BE + EF vs. 22 [20–24] in Recombinants *p* < 0.001. This trend reinforces the observation that the spike protein has a higher mutation rate compared to other viral proteins and shows its critical role in viral evolution and immune escape.

Moreover, the analysis of the number of SNPs with a mutated reads frequency ≥ 20% (cut-off corresponding to the Sanger method) was performed in the five main Omicron lineage groups. The analysis of non-synonymous SNPs ≥ 20% presented the same median [IQR] increase: 43 [41–44] in BA.1 vs. 50 [49–52] in BA.2 + BA.4 vs. 51 [49–52] in BA.5 + BF vs. 58 [57–59] in BQ + BE + EF vs. 70 [68–75] in Recombinants, *p* < 0.001 (Figure 4C). Similarly, the analysis of non-synonymous SNPs in Spike showed the median [IQR]: 27 [25–28] in BA.1 vs. 27 [27–28] in BA.2 + BA.4 vs. 29 [28–29] in BA.5 + BF vs. 31 [30–31] in BQ + BE + EF vs. 37 [31–38] in Recombinants, *p* < 0.001 (Figure 4D).

#### SNPs Differences Between IPs and NIPs, and Hospitalized and Non-Hospitalized Individuals

Analyzing potentially significant differences in the distribution of high-abundant SNPs between IPs and NIPs, we identified nine SNPs with a significantly different prevalence in the two groups. Seven SNPs were prevalently found in NIPs with respect to IPs, five of them located in the spike protein (G252V [nucleotide: G22317T] *p* = 0.020, G339H [nucleotide: G22577C] *p* = 0.020, G446S [nucleotide: G22898A] *p* = 0.036, N460K [nucleotide: T22942G] *p* = 0.009, and F490S [nucleotide: T23031C] *p* = 0.006), and the other two in the envelope protein (T11A [nucleotide: A26275G] *p* = 0.009) and in RdRp protein (G662S [nucleotide: A26275G] *p* = 0.009). Only two SNPs, both in the spike protein (I410I [nucleotide: C22792T] *p* = 0.022, and G339D [nucleotide: G22578A] *p* = 0.025) were prevalently found in IPs with respect to NIPs (Figure 5A).

Similarly, we observed significant differences when comparing the high-abundant SNPs between hospitalized individuals and non-hospitalized individuals. In detail, six SNPs were identified and all of them were prevalently found in non-hospitalized with respect to hospitalized individuals. Among them, one was localized in the spike protein (T376A [nucleotide: A22688G] *p* = 0.022), one in nucleocapsid protein (N8N [nucleotide: A22688G] *p* = 0.021), and four in non-structural proteins (nsp3:G1001S [nucleotide: G5720A] *p* = 0.020), nsp14:I42V [nucleotide: A18163G] *p* = 0.001, nsp15:T112I [nucleotide: C19955T] *p* = 0.007, and nsp15:E145E [nucleotide: A20055G] *p* = 0.006 (Figure 5B). Notably, the spike protein was the most involved with a total of seven SNPs with a significantly different distribution in IPs vs. NIPs, prevalently located in the RBD region (319aa–541aa). The observed mutational pattern between the two groups generally reflects the distribution of Omicron variants (Appendix A).

In addition, the number of high-abundant SNPs (frequency ≥ 40%), non-synonymous SNPs (frequency ≥ 20%), and low-abundant SNPs (frequency 5–19%) was analyzed to assess if there was a different evolution between IPs and NIPs, as well as between hospitalized and non-hospitalized individuals (Appendix A). No significant differences were observed in the median number of SNPs between IPs vs. NIPs, regardless of the SNPs category considered. Differently, a slightly higher number of low-abundant SNPs was observed in hospitalized vs. non-hospitalized individuals (median [IQR]: 2 [2–3] vs. 2 [1–2], *p* = 0.021) (Appendix A).

### 3.4. Covariation Profiles Among SARS-CoV-2 SNPs—Statistically Significant Pairs of SNPs

In our analysis of potential associations among SNPs, we found that 109 SNPs were involved in significant positive or negative associations in pairs. (Appendix A). Forty-seven SNPs were found in non-structural proteins, while the other sixty-two were found in structural and accessory proteins. Among these 109 SNPs, 107 showed positive associations in pairs (phi > 0.7 and *p*-value < 0.001) and 40 negative associations (phi < −0.7 and *p*-value < 0.001). The 43.9% (n = 47/107) of positive associations and the 52.5% (n = 21/40) of negative associations involved non-structural proteins, principally nsp3, nsp4, and RdRp proteins. In detail, most of the observed SNPs are located in spike (n = 43/109, 39.5%), among which 42 are involved in positive associations, followed by 10 in nsp3 (protease), and 6 in Nucleocapsid, nsp13 (helicase), and an RNA-dependent RNA polymerase, respectively.

### 3.5. Clusters of Correlated SNPs

Regarding the SNPs significantly associated in pairs, 19 SNPs were involved in 3 distinct clusters that exhibited positive correlations among them (Table 2). The first cluster (bootstrap = 0.99) includes three SNPs, with one of them located in the S protein (G23642T), and the other two located in N (G28936T) and RdRp (T15474G) proteins, respectively. The first two SNPs are non-synonymous, while the third was synonymous. These SNPs were detected with an intrapatient prevalence median [IQR] of 8.9 [8.0–9.9] for G23642T, 7.8 [6.7–9.2] for T15474G and 9.3 [6.8–11.6] for G28936T. The G28936T in the N gene and G23642T in the spike protein were also associated in the third cluster (bootstrap = 0.96) with the T23620G in the spike protein characterized by an intrapatient prevalence of 11.2 (10.2–12.8). Table 2 shows clusters of correlated SNPs.

The second cluster (bootstrap = 0.99) consisted of 13 SNPs mainly located in the spike protein and nsp3 (3 SNPs, including 1 synonymous in nsp3 and 6 non-synonymous SNPs in the spike protein). Two other synonymous SNPs were located in nsp10 and RdRp. In this cluster, the non-synonymous SNP A2832G in nsp3 showed an intrapatient prevalence median [IQR] of 88.1 [72.9–99.7] and was detected exclusively in the BA.1 lineage. All other SNPs (T5386G, G8393A, A11537G, T13195C, C15240T, C21846T, G23048A, C23202A, C24130A, C24503T, A26530G, and T22673C) were generally observed with very high intrapatient prevalence (>99.9).

Different frequencies in the five main Omicron lineages groups of the 19 SNPs involved in the 3 clusters were observed (Appendix A). No differences were found when analyzing the distribution of three clusters between the IPs and NIPs.

### 3.6. Correlation with Moderate/Severe COVID-19 Manifestation

Univariate and multivariate logistic regression models were conducted to determine whether immunocompromised status, hospitalization, or pneumonia related to COVID-19 could be associated with clusters of SNPs, lineages, and demographic characteristics (Table 3). Age, gender, previous vaccination, median CT values, and hematological comorbidities were included as confounding factors.

Univariate regression analysis revealed that hospitalization was positively associated with increased age and median CT values (OR [95% CI]: 1.02 [1.00–1.04], *p* = 0.028 and 1.14 [1.04–1.24], *p* = 0.004, respectively). In contrast, hospitalization was negatively associated with female gender and vaccination status (0.44 [0.23–0.84], *p* = 0.013 and 0.18 [0.07–0.48], *p* < 0.001, respectively).

Multivariate regression analysis showed a positive association between hospitalization and the cluster n. 3, characterized by the mutations Spike: S686R plus A694S and Nucleocapsid: L221F (AOR: 2.74 [95% CI: 1.13–6.64, *p* = 0.025]). Increased age was also positively associated with hospitalization (AOR:1.03 [95% CI: 1.00–1.06], *p* = 0.028). Conversely, negative associations were observed for female gender and previous vaccination status (AORs: 0.34 [95% CI: 0.14–0.83], *p* = 0.017 and 0.19 [95% CI: 0.06–0.63], *p* = 0.006, respectively). The results of regression analyses are reported in Table 3.

## 4. Discussion

The emergence of the Omicron variant of SARS-CoV-2 profoundly influenced the COVID-19 pandemic. The Omicron variant is highly contagious and easily transmissible, but it generally leads to less severe illness compared to earlier variants. In our retrospective analysis, we highlighted the complex epidemiologic dynamics of Omicron and its different sublineages in Central Italy during its first two years of circulation (December 2021–December 2023), with a particular focus on hospitalizations and the impact and progression of the disease in immunocompromised individuals. Most individuals were non-hospitalized (n = 259, 84.6%), and a significantly higher frequency of vaccination was observed in non-hospitalized (95.3% *p* < 0.001), suggesting a potential protective role of vaccination in preventing severe outcomes. Among hospitalized people, a higher rate of pneumonia was observed in NIPs compared to IPs (85.7% vs. 34.6%; *p* = 0.002). These data likely suggest that IPs were more closely monitored, and second, they required hospitalization for conditions other than pneumonia, differently from NIPs. Additionally, IPs were younger than NIPs [60 (47–71) vs. 73 (60–81) years; *p* < 0.001, respectively].

The study shows that the Omicron variant was the dominant circulating variant in Central Italy, as well as globally, from December 2021 to December 2023. The analysis of 306 SARS-CoV-2 whole-genome sequences revealed notable viral evolution. A refined phylogenetic analysis of the genomic sequences identified at least five main Omicron lineage groups. Most SARS-CoV-2 infections were attributed to the BA.1 group (34%), followed by the BA.2 + BA.4 group (25.8%), Recombinant forms (20.2%), the BA.5 + BF group (10.8%), and the BQ + BE + EF group collectively accounting for 9.2% of infections. Interestingly, a different prevalence of five main lineages was observed between IPs and NIPs: Recombinant lineages were less prevalent in IPs (16.0%) compared to NIPs (27.1%; *p* = 0.018), while the BA.2 + BA.4 lineages were more common in IPs (30.9%) than in NIPs (17.8%; *p* = 0.011). When evaluating the population based on hospitalization and immunocompromised status, the analysis revealed that BA.2 + BA.4 lineages were significantly more prevalent in the non-hospitalized IPs (NHIP) group (33.3%) compared to the other groups (15.4% in HIP, 14.3% in HP, and 18.6% in NHP, *p* = 0.001). In contrast, the Omicron BA.1 group was more prevalent in hospitalized non-IPs (HP group) (57.1%) compared to the others (34.6% in HIP, 32.1% in NHIP, and 32.0% in NHP; *p* = 0.020). An in-depth characterization of single nucleotide polymorphisms was conducted to evaluate the mutation rate across the five main Omicron groups during its spread. Interestingly, the continuous evolution of the main SARS-CoV-2 Omicron lineages reflects the trend of SNP prevalence across emerging lineages (from BA.1 to the Recombinant lineages). Overall, we observed a significant increase in high-abundance SNPs (≥40% read frequency), rising from 54 SNPs in BA.1 to 92 SNPs in Recombinant lineages. This trend is particularly evident in the spike protein, which increased from 28 SNPs in BA.1 to 39 in the Recombinants (*p* < 0.001), as well as in the receptor-binding domain, which increased from 15 SNPs in BA.1 to 22 in the Recombinant lineages. Furthermore, non-synonymous SNPs with a read frequency ≥ 20% also showed a notable significant increase (from 43 in BA.1 to 70 in Recombinant lineages [*p* < 0.001]). As previously reported, the accumulation of these mutations defines new lineages known to exhibit increased infectivity, transmission, and immune escape [1,24]. The spike protein followed the same increase, showing from 27 non-synonymous SNPs in BA.1 to 37 in the Recombinant lineages (*p* < 0.001), highlighting its critical role in the evolution of SARS-CoV-2. Notably, these mutations contribute to the enhanced ability of the latest Omicron clades to evade the immune response, even in vaccinated individuals [25], although with a reduction in the severity of COVID-19 [26]. The non-synonymous SNPs trend reinforces the hypothesis that these mutations could provide functional advantages to the virus. Furthermore, some SNPs were involved in mutational clusters. In detail, the first cluster identified involved three low-abundant SNPs: spike-A694S [nucleotide: G23642T], RdRp-G678G [nucleotide: T15474G], and Nucleocapsid-L221F [nucleotide: G28936T]. These SNPs presented different prevalences across the main five Omicron lineage groups, with a higher frequency in the BA.2 + BA.4 group (>50%). Notably, the RdRp-G678G was not detected in Recombinant lineages. The second cluster included 13 SNPs, primarily found in the Spike and nsp3 proteins. The SNP nsp3-K38R [nucleotide: A2832G], detected with a median intrapatient prevalence of 88.1% [72.9–99.7], was associated with constitutive SNPs mainly detected in the BA.1 group. The third cluster, found in all five main Omicron lineages, was composed of the spike mutations S686R [nucleotide: T23620G] and A694S [nucleotide: G23642T] with Nucleocapsid- L221F [nucleotide: G28936T]. A previous study showed that the low-abundant S686R mutation was associated with the worst clinical manifestation of COVID-19 [27].

To further investigate, additional analyses were performed to explore significant associations among demographic factors, vaccination status, specific clusters of mutations, and the severity of COVID-19 outcomes. The univariate analysis showed that advanced age and median CT values are associated with an increased likelihood of hospitalization (OR [95% CI]: 1.02 [1.00–1.04], *p* = 0.028 and 1.14 [1.04–1.24], *p* = 0.004, respectively), potentially reflecting a combination of host immune senescence and higher viral load. In contrast, female gender and prior vaccination are protective factors against hospitalization (OR [95% CI]: 0.44 [0.23–0.84], *p* = 0.013 and 0.18 [0.07–0.48], *p* < 0.001, respectively), suggesting both sex-based immunological differences and the effectiveness of vaccination in reducing disease severity [28,29]. The multivariate analysis emphasizes the role of a specific mutation cluster in the Spike and Nucleocapsid proteins (Spike: S686R and A694S; Nucleocapsid: L221F) as a predictor of hospitalization. An adjusted odds ratio (AOR) of 2.74 for this cluster suggests a substantial impact on clinical outcomes, likely enhancing the viral ability to escape host immune responses or to interact with host cells more effectively. The furin cleavage site (^680^SPRRARSVAS^689^) in the S glycoprotein is important for the pathogenesis and evolution of SARS-CoV-2. Changes in this region impact immune evasion and reduce vaccine efficacy [30]. This cluster, alongside age, is a positive predictor of hospitalization, underscoring the need for close monitoring of these mutations as part of genomic surveillance activities. In our analysis, the BQ + BE + EF group was positively associated with hospitalization. A slightly increased frequency of BQ + BE + EF lineages was observed in the hospitalized IPs (HIP group), accounting for 19.2%, compared to 8.0% in NHIP, 9.5% in HP, and 8.2% in NHP (*p* = 0.06). This suggests that this group may be associated with more severe outcomes, although evidence is only available for the BQ.1 Omicron lineage, which shows a higher frequency of hospitalization [31,32]. Furthermore, BQ presents several mutations that could potentially enhance immune evasion, thus increasing the risk of more severe outcomes in vulnerable individuals. This leads to greater persistence of infection in these individuals, potentially facilitating the emergence of new variants [33].

## 5. Conclusions

In conclusion, our findings showed the evolution of the SARS-CoV-2 Omicron lineages in Central Italy. Analyses revealed that hospitalization was associated with a specific group (BQ + BE + EF lineages), a specific cluster of mutations (which S686R involved in spike protein), and increasing age, while female sex and prior vaccination provided protection. A notable increase in SNPs from BA.1 to Recombinant lineage groups, particularly in the Spike protein, highlights how the trend of the SNPs’ prevalence across the lineages well represented the continuous evolution of SARS-CoV-2. These results enhance the understanding of Omicron variability in both immunocompromised and non-immunocompromised individuals, emphasizing the importance of continuous genomic surveillance and targeted health strategies for vulnerable populations to prevent SARS-CoV-2-related disease.

## Figures and Tables

**Figure 1 viruses-17-00540-f001:**
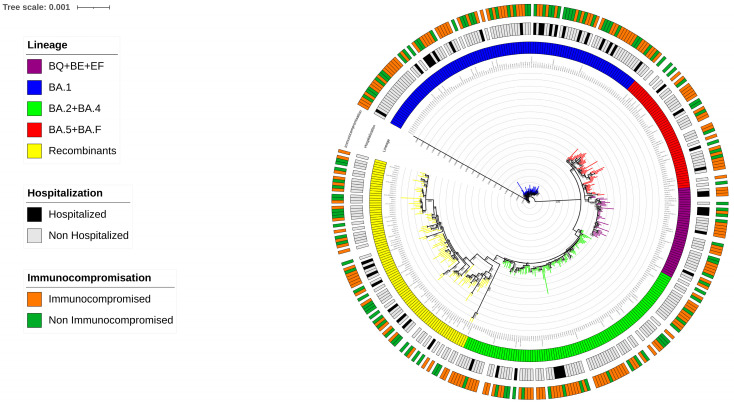
The Maximum Likelihood phylogenetic tree shows the distribution of the 306 SARS-CoV-2 sequences analyzed in relation to the clinical characteristics of individuals, with main Omicron lineages represented by a different color. The phylogenetic tree highlights the prevalence of at least five major Omicron lineage groups and their association with clinical outcomes, including immunocompromised status and hospitalization rates, both indicated with specific colors within the tree.

**Figure 2 viruses-17-00540-f002:**
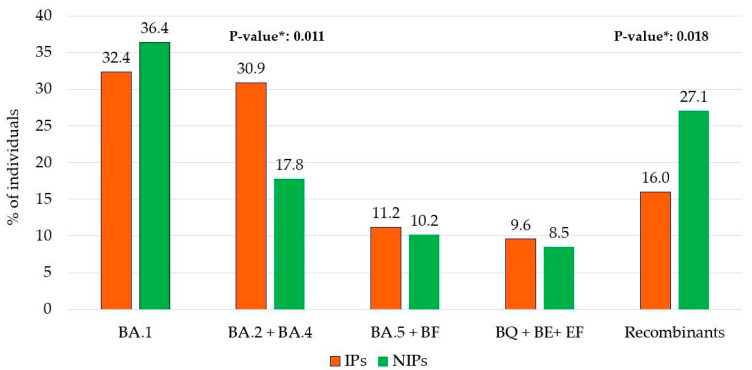
Distribution of immunocompromised (IPs) and non-immunocompromised (NIPs) individuals within the 5 main Omicron lineage groups. * *p*-values were calculated by Chi-squared test comparing IPs vs. NIPs within each group.

**Figure 3 viruses-17-00540-f003:**
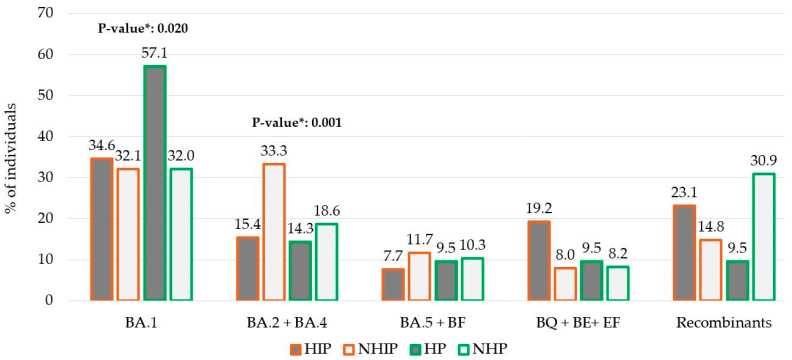
Distribution of four groups of individuals within the 5 main Omicron group lineages according to immunocompromised and/or hospitalization status. Abbreviations: HIP: Hospitalized immunocompromised People; NHIP: Non-Hospitalized Immunocompromised People; HP: Hospitalized non-immunocompromised People; NHP: Non-Hospitalized and non-immunocompromised People. * *p*-values were calculated by Chi-squared test comparison in BA.1 **HP** vs. the other groups; in BA.2 + BA.4 **NHIP** vs. the other groups; and in BQ + BE + EF **HIP** vs. the other groups.

**Figure 4 viruses-17-00540-f004:**
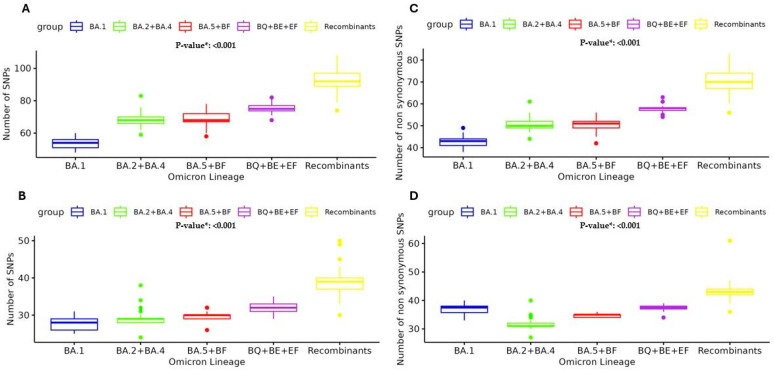
Median number and Interquartile range of high-abundant SNPs (frequency ≥ 40%) observed against main Omicron lineage groups: in whole genome (**A**) and spike (**B**); non-synonymous-abundant SNPs (frequency ≥ 20%) observed in whole genome (**C**) and spike (**D**). SNPs single nucleotide polymorphism. * *p*-values were calculated by the Kruskal–Wallis test.

**Figure 5 viruses-17-00540-f005:**
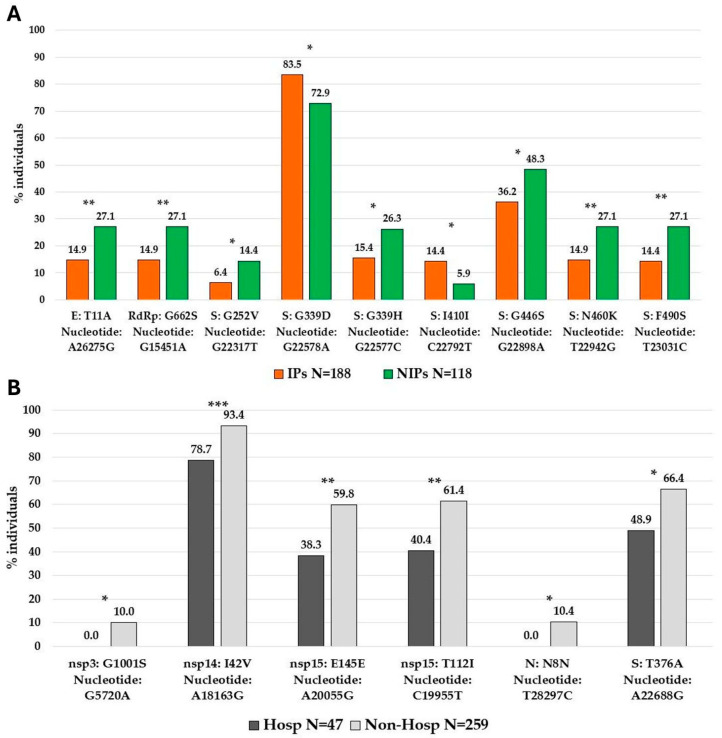
SNPs differences observed between IPs and NIPs, and Hospitalized and Non-Hospitalized individuals. Prevalences of 9 high-abundant SNPs found in IPs vs. NIPs (**A**) and of 6 high-abundant SNPs in hospitalized vs. non-hospitalized individuals (**B**). SNPs single nucleotide polymorphism. Prevalences are expressed as N (%). *p*-values were calculated by Chi-square test or Fisher’s exact test, as appropriate, and indicated by stars * *p* < 0.05; ** *p* < 0.01; *** *p* < 0.001.

**Table 1 viruses-17-00540-t001:** Characteristics of the Study Population.

Variables	Overall N = 306	IPs N = 188		NIPs N = 118		*p*-Value *^a^*		
		Hospitalized (HIP) N = 26	Non-Hospitalized (NHIP) N = 162	Hospitalized (HP) N = 21	Non-Hospitalized (NHP) N = 97	IPs vs. Non-IPs	HIP vs. NHIP	Hosp vs. Non-Hosp
**Hospitalized, n (%)**	47 (15.4)	26 (13.8)	-	21 (17.8)	-			
**Non-Hospitalized, n (%)**	259 (84.6)	-	162 (86.2)	-	97 (82.2)			
**Female, n (%)**	156 (50.9)	7 (26.9)	91 (56.2)	9 (42.9)	49 (50.5)	0.612	**0.006**	**0.012**
**Age, years, median (IQR)**	64 (51–76)	68 (50–76)	58 (45–69)	73 (65–79)	73 (58–81)	**<0.0001**	0.0629	**0.026**
**Month of sample collection, median (IQR)**	Apr 2022 (Feb 2022–Dec 2022)	May 2022 (Feb 2022–Jan 2023)	Apr 2022 (Feb 2022–Oct 2022)	Feb 2022 (Feb 2022–Oct 2022)	Jun 2022 (Feb 2022–Apr 2023)	0.2005	0.7263	0.2801
**Pneumonia, n (%)**	27 (8.8)	9 (34.6)	0 (0.0)	18 (85.7)	0 (0.0)	**0.002**	-	-
**Immunocompromised, n (%)**	188 (61.4)	26 (100)	162 (100)	-	-	-	-	-
**Cycle-Threshold (Ct):**								
E, median (IQR)	21 (19–24)	23 (21–26)	21 (19–24)	23 (18–31)	21 (19–23)	0.896	0.374	**0.035**
N, median (IQR)	20 (18–23)	22 (20–25)	20 (18–23)	23 (17–30)	20 (18–23)	0.787	0.441	0.121
RdRp/S, median (IQR)	23 (20–25)	24 (23–27)	23 (20–25)	22 (18–29)	22 (20–25)	0.968	0.226	**0.038**
**Vaccinated *^b^*, n (%)**	276 (92.9)	18 (85.7)	155 (95.7)	12 (66.7)	91 (95.8)	0.171	0.059	**<0.0001**

Abbreviations: IPs: Immunocompromised adult People; NIPs: Non-Immunocompromised adult People; HIP: Hospitalized Immunocompromised People; NHIP: Non-Hospitalized Immunocompromised People; HP: Hospitalized non-immunocompromised People; NHP: Non-Hospitalized and non-immunocompromised People. IQR: interquartile range. Data are expressed as median (IQR) or N (%). *^a^ p*-values were calculated by Mann-Whitney, Kruskal-Wallis and Chi Square tests as appropriate, and statistically significant values are shown in bold. *^b^* Data available for 297 individuals: HIP n = 21, NHIP n = 162, HP n = 18 and NHP n = 96.

**Table 2 viruses-17-00540-t002:** Covariation Profiles Among SARS-CoV-2 SNPs.

Cluster	SNP 1	Amino Acid Residue	Prevalence n (%)	Intrapatient Prevalence Median (IQR)	SNP 2	Amino Acid Residue	Prevalence n (%)	Intrapatient Prevalence Median (IQR)	Covariation Freq n (%) ^a^	Covariation Freq n (%) ^b^	Phi ^c^	*p*-Value ^d^
**Cluster 1**	G23642T	S: A694S	145 (47.4)	8.9 (8.0–9.9)	T15474G	RdRp: G678G	107 (35.0)	7.8 (6.7–9.2)	102/145 (70.3)	102/107 (95.3)	0.70	2.23 × 10^−37^
					G28936T	N: L221F	131 (42.8)	9.3 (6.8–11.6)	115/145 (79.3)	115/131 (87.8)	0.70	2.51 × 10^−36^
**Cluster 2**	A2832G	nsp3: K38R	104 (34.0)	88.1 (72.9–99.7)	T5386G	nsp3: A889A	104 (34.0)	99.9 (99.9–100)	104/104 (100)	104/104 (100)	1.00	9.82 × 10^–83^
					G8393A	nsp3: A1892T	107 (35.0)	99.8 (99.7–99.9)	104/104 (100)	104/107 (97.2)	0.98	7.77 × 10^–78^
					A11537G	nsp6: I189V	104 (34.0)	99.9 (99.9–99.9)	104/104 (100)	104/104 (100)	1.00	9.82 × 10^–83^
					T13195C	nsp10: V57V	104 (34.0)	99.9 (99.9–99.9)	104/104 (100)	104/104 (100)	1.00	9.82 × 10^–83^
					C15240T	RdRp: H599H	105 (34.3)	99.9 (99.9–100)	104/104 (100)	104/105 (99.0)	0.99	6.11 × 10^–81^
					C21846T	S: T95I	104 (34.0)	99.9 (99.9–100)	104/104 (100)	104/104 (100)	1.00	9.82 × 10^–83^
					G23048A	S: G496S	105 (34.3)	99.9 (99.8–100)	104/104 (100)	104/105 (99.0)	0.99	6.11 × 10^–81^
					C23202A	S: T547K	107 (35.0)	99.9 (99.9–100)	104/104 (100)	104/107 (97.2)	0.98	2.66 × 10^–79^
					C24130A	S: N856K	104 (34.0)	99.9 (99.9–100)	104/104 (100)	104/104 (100)	1.00	9.82 × 10^–83^
					C24503T	S: L981F	104 (34.0)	99.9 (99.9–99.9)	104/104 (100)	104/104 (100)	1.00	9.82 × 10^–83^
					A26530G	M: D3G	103 (33.7)	99.9 (99.9–99.9)	102/104 (98.1)	102/103 (99.0)	0.98	8.90 × 10^–79^
					T22673C	S: S371L	103 (33.7)	100 (99.9–100)	103/104 (99.0)	103/103 (100)	0.99	1.11 × 10^–80^
**Cluster 3**	T23620G	S: S686R	193 (63.1)	11.2 (10.2–12.8)	G23642T	S: A694S	145 (47.4)	8.9 (8.0–9.9)	139/193 (72.0)	139/145 (95.9)	0.64	2.28 × 10^−32^
					G28936T	N: L221F	131 (42.8)	9.3 (6.8–11.6)	128/193 (66.3)	128/131 (97.7)	0.62	2.54 × 10^−31^

SNP single nucleotide polymorphism. IQR interquartile range. ^a^ Covariation frequency based on the prevalence of SNP 1. ^b^ Covariation frequency based on the prevalence of SNP 2. ^c^ Positive correlations with phi > 0.7. ^d^ All p-values for covariation were significant at a false discovery rate of 0.05.

**Table 3 viruses-17-00540-t003:** Multivariate Logistic Regression Analysis of Factors Associated with Hospitalization.

Variable Associated to Hospitalization	Univariate Analysis		Multivariate Analysis	
	OR (95% CI)	*p*-Value	AOR (95% CI)	*p*-Value
**Gender (female vs. male)**	0.44 (0.23–0.84)	**0.013**	0.34 (0.14–0.83)	**0.017**
**Age (per one year increase)**	1.02 (1.00–1.04)	**0.028**	1.03 (1.00–1.06)	**0.028**
**Immunological diseases (yes vs. no)**	0.74 (0.40–1.39)	0.35		
**Vaccination status (yes vs. no)**	0.18 (0.07–0.48)	**<0.001**	0.19 (0.06–0.63)	**0.006**
**SARS-CoV-2 group**				
**BA.1**	1.71 (0.91–3.22)	0.095		
**BA.2 + BA.4**	0.45 (0.19–1.06)	0.068		
**BA.5 + BF**	0.74 (0.25–2.20)	0.568		
**BQ + BE + EF**	1.98 (0.79–4.97)	0.144	4.00 (1.16–13.75)	**0.028**
**Recombinants**	0.78 (0.34–1.76)	0.549		
**Ct mean value**	1.14 (1.04–1.24)	**0.004**		
**Clusters**				
**1**	1.32 (0.69–2.51)	0.4		
**2**	0.94 (0.47–1.88)	0.856		
**3**	1.15 (0.61–2.17)	0.662	2.74 (1.13–6.64)	**0.025**

Univariate and multivariate logistic regression analysis of factor associated with hospitalization. Abbreviations: CI: confidence interval; OR: odds ratio; AOR: adjusted odds ratio. Ct: Cycle threshold. Statistically significant *p*-values are shown in bold.

## Data Availability

SARS-CoV-2 sequences were submitted to GISAID.

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
