# Peer review of "An In-Depth Characterization of SARS-CoV-2 Omicron Lineages and Clinical Presentation in Adult Population Distinguished by Immune Status"

_viruses, 2025, doi:10.3390/v17040540_

Round 1
Reviewer 1 Report
Comments and Suggestions for Authors
Reviewer Comments
Title: An in-depth characterization of SARS-CoV-2 Omicron Lineages and clinical presentation in adult population distinguished by immune status
Journal: Viruses
The authors performed the full genome analysis of SARS-CoV-2 from 306 samples with confirmed cases of COVID-19 infection collected during December 2021 and December 2023 in Central Italy with the aim to provide a retrospective analysis of variability of SARS-CoV-2 Omicron variant during its spread over first two years after the emergence of Omicron variants. The study introduced the interesting evidence; however, I would like to provide the comment as below.
- This study focuses on both the characterization of Omicron lineages and the clinical presentation in immunosuppressed (IPs) and non-immunosuppressed patients (NIPs). While the distribution and frequency of Omicron variants across five groups were compared between IPs and NIPs, a detailed comparison of clinical presentation, hospitalization, and mutation patterns between these two groups was not provided which was not in align with the title of the manuscript. Additionally, a multiple logistic regression model should be performed based on immune status.
- Regarding clinical presentation, the variables included in the analysis are very limited. The authors should include additional factors such as hospitalization duration, disease severity, medications, and other relevant blood test results, including anti-Spike IgG, anti-N IgG, and neutralizing antibodies if data are available.
Author Response
1) Review
Thank you very much for taking the time to review this manuscript. Please find below our detailed responses, along with the corresponding revisions and corrections, which have been highlighted in track changes in the re-submitted files.
Comment 1: This study focuses on both the characterization of Omicron lineages and the clinical presentation in immunosuppressed (IPs) and non-immunosuppressed patients (NIPs). While the distribution and frequency of Omicron variants across five groups were compared between IPs and NIPs, a detailed comparison of clinical presentation, hospitalization, and mutation patterns between these two groups was not provided which was not in align with the title of the manuscript.
Response: As correctly suggested, the manuscript now includes a section dedicated to the comparison of the mutational pattern, with a subsection titled 3.3.1 “SNPs differences between IPs and NIPs, and Hospitalized and Non-Hospitalized Individuals,' in which mutations exhibiting a significantly different distribution between IPs and NIPs, as well as between hospitalized and non-hospitalized individuals, are presented in pages 8-10 lines 310-354.
Furthermore, a two-panel figure has been added to the manuscript, showing mutations (high-abundant synonymous and non-synonymous SNPs) identified with a significantly different distribution between IPs and NIPs, as well as between hospitalized and non-hospitalized individuals. Additionally, in the supplementary materials, we have included: a) a table presenting the specific frequencies of these mutations, both between the groups and across the 5 Omicron clades; b) a figure containing the median number and interquartile range of high-abundant SNPs (frequency ≥40%), non-synonymous SNPs (frequency ≥20%), and low-abundant SNPs (frequency 5-19%) observed in IPs vs NIPS and in hospitalized vs non-hospitalized.
The immune status was already added in our multivariate logistic regression analysis as factor potentially associated with hospitalization. Regarding clinical presentation, the differences in the frequency of pneumonia are reported in Table 1. We have emphasized this result in the text of paragraph 3.1, with the following new sentence on page 4, lines 184-187:
Pneumonia was observed exclusively in hospitalized individuals. When comparing hospitalized IPs vs NIPs individuals, the prevalence of pneumonia was found to be significantly lower in hospitalized IPs compared to NIPs (34.6% in HIP vs 85.7% in HP, p=0.002).
Comment 2: Regarding clinical presentation, the variables included in the analysis are very limited. The authors should include additional factors such as hospitalization duration, disease severity, medications, and other relevant blood test results, including anti-Spike IgG, anti-N IgG, and neutralizing antibodies if data are available.
Response: Thank you for pointing this out. Regarding the clinical presentation of the 306 individuals in our study, all available variables were reported. The analysis was focused on the data at our disposal, maximizing the use of the information available.

Reviewer 2 Report
Comments and Suggestions for Authors
Thank you very much to the Editor of Viruses for allowing me to review the paper entitled “An in-depth characterization of SARS-CoV-2 Omicron lineages and clinical presentation in adult population distinguished by immune status”, purposed by Bellocchi MC and colleagues.
The authors have submitted an interesting retrospective analysis of the variability of the SARS-CoV-2 Omicron variant during its spread from December 2021 to December 2023 in Central Italy. In this communication the authors highlight the importance of continuous genomic surveillance and targeted health strategies for vulnerable populations to prevent SARS-CoV-2 related disease. The paper is based on rich and new literature, it is well structured, the figures and tables are easy to understand for the reader and the discussion are complete for the data shown.
It’s my opinion that this paper deserves publication. Some comments to be addressed by the authors:
- Table 1 shown a significant different between Hosp vs Not-Hosp as regard the Ct value. Please insert it in the results, being a result that will be discussed later
- Line 398: The authors discussed that their results confirm the protective role of vaccination. It’s my opinion that the authors should consider it as a probability, since numerous studies have shown that the Omicron variant evaded the neutralizing power of vaccines made with different viral strains.
Author Response
2) Review
Thank you very much for taking the time to review this manuscript. Please find below our detailed responses, along with the corresponding revisions and corrections, which have been highlighted in track changes in the re-submitted files.
Comments 1: Table 1 shown a significant different between Hosp vs Non-Hosp as regard the Ct value. Please insert it in the results, being a result that will be discussed later
Response 1: In agreement with the reviewer’s comment, we have modified the results section by inserting the sentence on page 4, lines 188-190:
“In addition, a significant difference was observed when comparing the median Ct values between hospitalized and non-hospitalized individuals, particularly for the E and RdRp/S targets (p=0.035 and p=0.038, respectively)”
Comments 2: Line 398: The authors discussed that their results confirm the protective role of vaccination. It’s my opinion that the authors should consider it as a probability, since numerous studies have shown that the Omicron variant evaded the neutralizing power of vaccines made with different viral strains.
Response 2: Thank you for pointing this out. I have changed the term 'confirming' to 'suggesting' in the sentence on page 12, lines 440-441:
“suggesting a potential protective role of vaccination in preventing severe outcomes”
